# Comparison of Antibiofilm Activity of *Pseudomonas aeruginosa* Phages on Isolates from Wounds of Diabetic and Non-Diabetic Patients

**DOI:** 10.3390/microorganisms11092230

**Published:** 2023-09-04

**Authors:** Sarika Suresh, Joylin Saldanha, Ashwini Bhaskar Shetty, Ramya Premanath, D. S. Akhila, Juliet Roshini Mohan Raj

**Affiliations:** Division of Infectious Diseases, Nitte University Center for Science Education and Research, Paneer Campus, Nitte (Deemed to be University), Derelakatte, Mangaluru 575018, Indiaramya@nitte.edu.in (R.P.); akhila@nitte.edu.in (D.S.A.)

**Keywords:** bacteriophages, biofilm removal, *Pseudomonas aeruginosa*, wounds

## Abstract

The persistence of organisms as biofilms and the increase in antimicrobial resistance has raised the need for alternative strategies. The study objective was to compare the ability of isolated bacteriophages to remove in vitro biofilms formed by *Pseudomonas aeruginosa* isolated from the environment with those isolated from diabetic and non-diabetic wounds. *P. aeruginosa* were isolated from clinical and environmental sites, and antimicrobial susceptibility was tested. Bacteriophages were isolated and characterized based on plaque morphology and host range. A reduction in the viable count assayed the lytic ability of candidate phages. The crystal violet method was used to determine the residual biofilm after 24 h of phage treatment on 72-h-old biofilms. The statistical significance of phage treatment was tested by one-way ANOVA. Of 35 clinical isolates, 17 showed resistance to 1 antibiotic at least, and 7 were multidrug resistant. Nineteen environmental isolates and 11 clinical isolates were drug-sensitive. Nine phages showed 91.2% host coverage, including multidrug-resistant isolates. Phages eradicated 85% of biofilms formed by environmental isolates compared to 58% of biofilms of diabetic isolates and 56% of biofilms of non-diabetic isolates. Clinical isolates are susceptible to phage infection in planktonic form. Biofilms of *P. aeruginosa* isolated from diabetic wounds and non-diabetic wounds resist removal by phages compared to biofilms formed by environmental isolates. All phages were efficient in dispersing PAO1 biofilms. However, there was a significant difference in their ability to disperse PAO1 biofilms across the different surfaces tested. Partial eradication of biofilm by phages can aid in complementing antibiotics that are unable to penetrate biofilms in a clinical set-up.

## 1. Introduction

*Pseudomonas aeruginosa* is a notorious opportunistic pathogen known to affect debilitated patients, particularly in hospitals. It is an important nosocomial pathogen associated with burn wounds, surgical wounds, diabetic foot ulcers, and bedsores [1]. To survive and establish itself within the host, it produces a vast arsenal of virulence factors, including the production of exopolysaccharides, phospholipases, exoproteases, type III effectors, and phenazines [2]. *Pseudomonas aeruginosa* is commonly associated with wound infections and is thrice more frequent in people with diabetes than non-diabetics. People with diabetes have a lifetime risk of developing diabetic foot ulcers caused by various Gram-positive and Gram-negative bacteria. *P. aeruginosa* is the most frequently encountered Gram-negative bacteria in diabetic foot ulcers leading to increased morbidity and mortality [3]. Antimicrobial resistance (AMR) in *P. aeruginosa,* especially carbapenem resistance, has emerged as a global outbreak. The ability of this bacterium to resist many of the currently available antibiotics has made the treatment of *P. aeruginosa* infections a significant challenge. The innate immunity of *P. aeruginosa* toward many classes of antimicrobials and its versatile ability to take up genes has made it a World Health Organization priority to develop new and alternative strategies [4]. *P. aeruginosa* is known to produce biofilms in the hospital environment, providing additional protection against antibiotics and a powerful capability of evading host defenses [5]. Thus, alternative feasible strategies to combat bacterial infections, especially those that form biofilms, are needed. Phage therapy is one such therapy where viruses effectively destroy specific bacterial cells [6]. Successful treatment using specific phages to treat *Pseudomonas* infections has been reported [7,8,9,10,11]. Bacteriophages are natural predators of bacteria. Phages prevent the destruction of skin grafts by *P. aeruginosa* in guinea pigs [12]. However, reports of phages have shown promising results in vitro but failed to deliver the same in vivo. This difference is because phage efficacy is often tested only on planktonic cells, but in vivo, organisms form biofilms that provide better survival [13,14]. The ability of *P. aeruginosa* to form biofilms correlates with its ability to impair the healing of diabetic wounds [15]. Recent studies indicate that the geographic region, type of chronic wound infection, wound size, duration of the injury, and use of active dressing influence the probability and severity of *P. aeruginosa* infections [16]. On the contrary, unlike other pathogenic bacteria, clinical and environmental isolates of *P. aeruginosa* show little genetic variability. Its low genetic diversity could be attributed to the fact that the clinical strains constitute a subpopulation present in environments close to human populations with the ability to produce virulence-associated traits [17]. As the environmental isolates of *P. aeruginosa* have the potential to produce virulence factors, they are considered potential pathogens [18]. Phages are locally adapted to their bacterial hosts but tend to increase their host range during the initial stages of coevolution [19]. With this background, this study was initiated to compare the biofilm-forming ability of *P. aeruginosa* isolates from diabetic and non-diabetic wounds and further compare the ability of bacteriophages to reduce *Pseudomonas* biofilms formed by isolates from different sources. 

## 2. Materials and Methods

### 2.1. Bacterial Isolates

Clinical samples (wound swabs) from patients of a tertiary care hospital in Mangaluru, India, were obtained on approval from the Nitte (Deemed to be University) Institutional Ethics Committee (INST.EC/2017-18/003). Patients consented before sample collection. Swabs of infected wounds (n = 67) of patients with a history of diabetes (diabetic wound isolates DW) and those without a history of diabetes (non-diabetic wound isolates NDW) were inoculated into Asparagine broth as the enrichment media and incubated at 37 °C for 24 h. Isolates were selected by culturing on cetrimide agar and identified based on biochemical tests and further confirmed as *P. aeruginosa* by polymerase chain reaction (PCR) for *oprL* [20]. A total of 15 freshwater sources and 15 soil samples from various locations in Mangaluru, India, were processed similarly to isolate *P. aeruginosa* and included as a non-clinical environmental group for comparison. The antibiotic susceptibility was determined by the disc diffusion method as per CLSI guidelines 2019, and *P. aeruginosa* ATCC 27853 was maintained as the quality control strain. Antibiotic discs of Cefoperazone (75 mcg), Piperacillin (100 mcg), Levofloxacin (5 mcg), Gentamicin (10 mcg), Amikacin (30 mcg), Imipenem (10 mcg), Aztreonam (30 mcg), Piperacillin/Tazobactam (100/10 mcg), Ceftazidime (30 mcg), Netillin (30 mcg), Ciprofloxacin (5 mcg), and Tobramycin (10 mcg) (from HiMedia, Mumbai, India) were used. The zones of inhibition (mm) after an incubation period of 24 h were measured to interpret susceptibility.

### 2.2. Bacteriophages

*Pseudomonas* phages were isolated from the same water and soil samples for bacteria isolation. Phages were isolated by baiting the strain PAO1 by the soft agar overlay method. Briefly, samples were sedimented at 4800× *g*, 4 °C for 20 min; the supernatant was filtered through a 0.22 µm syringe filter and then used as the phage source. A total of 1 mL of the host culture PAO1 was mixed with 1 mL of the phage source, overlaid with soft agar on a nutrient agar plate and incubated at 37 °C for 18 h. Lysates were purified by three consecutive rounds of single plaque purification. Fourteen plaques were successfully propagated, as verified by the routine test dilution wherein 5 µL of phage lysate dilutions were spotted on a lawn of PAO1 and observed after 24 h of incubation at 37 °C. A total of 50 mL of each phage lysate were purified by ultracentrifugation at 32,000× *g* for 4 h. The pellet was suspended in 500 µL SM buffer (50 mM Tris-HCl pH7. 5, 100 mM NaCl, 8 mM MgSO_4_, 0.01% Gelatin). These purified phages were diluted in the SM buffer as per the required number of phages and used for all further studies. To determine the structure of the candidate phages, transmission electron microscopy of the selected phages was performed at the Department of Metallurgy, National Institute of Technology Karnataka, Surathkal. Ten microlitres of high-titer purified phage lysate were stained with 3% neutral phosphotungstic acid and mounted onto 200 mesh carbon-coated copper grids for visualization at 120 Kilovolts in a JOEL (Tokyo, Japan) JEM 2100 electron microscope. 

### 2.3. Lytic Activity of Phages

The spot assay determined the host range or spectrum of the phages on all the clinical and environmental isolates. The viable count reduction assay determined the lytic ability of three candidate phages (chosen based on host spectrum and plaque morphology) [21]. The bacterial culture was grown in nutrient broth to OD_600_ of 0.5 and then infected with phages at the multiplicity of infection (MOI) 10, 1, 0.1, 0.01, and 0.001. Viable cell counts were determined by enumerating the colonies formed on 100 µL of plating of the culture (sampled every 2 h, up to 16 h) on nutrient agar. The statistical significance of the ability of phages to reduce viable counts was tested by one-way ANOVA. In combinations where the phage caused an initial decrease in the viable count followed by an increase, colonies that appeared were cultured on nutrient agar. The cultures that grew after two consecutive rounds of sub-culturing were tested for phage susceptibility by the spot assay to identify phage-resistant mutants if any [22]. 

### 2.4. Antibiofilm Activity of Phages

*Pseudomonas* cultures were grown overnight and diluted in 1:50 nutrient broth. One hundred microlitres of each dilute culture were loaded in quadrate onto a 96-well polystyrene microtiter plate and incubated at 37 °C for 72 h. Cultures were aspirated out to remove planktonic cells and air-dried. A total of 100 µL of phage (1 × 10^5^ PFU/mL) was added into the wells. One set was maintained without phage as the cell control. The plates were incubated at 37 °C overnight. Crystal violet biofilm assay was performed [23]. 

The following formula was used to compare the biofilm formation and dispersion achieved:$$\mathrm{Percentage}\;\mathrm{biofilm}\;\mathrm{formed}\;\mathrm{after}\;\mathrm{phage}\;\mathrm{treatment}\;\;\;\;\;\;\;\;\;\;\;\;=\frac{\mathrm{Absorbance}\;\mathrm{of}\;\mathrm{phage}\;\mathrm{treatment}}{\mathrm{Absorbance}\;\mathrm{of}\;\mathrm{cell}\;\mathrm{control}}\times 100\%$$

### 2.5. Activity of Phages on Biofilms Formed on Different Materials

The biofilm formation of PAO1 on three abiotic surfaces, i.e., polystyrene (microtiter plate), glass, and stainless steel (SS), was compared, and the effectiveness of the phages in disrupting these biofilms was determined. All tests were performed for three biological replicates.

Polystyrene: biofilm formation in the PES microtiter plate was performed as mentioned in Section 2.4.

Glass: 27 mL capacity glass tubes were inoculated with 5 mL of a 1:50 dilute culture and incubated at 37 °C for 72 h. The culture was aspirated out, and the tubes were let to dry. Two mL of phage (1 × 10^5^ PFU/mL) were added. Care was taken to add all solutions without touching the upper culture ring of the culture-air interface. The test tubes were incubated at 37 °C overnight. Crystal violet biofilm assay was performed as per O’Toole’s procedure, adding 2 mL volumes of 0.1% crystal violet and acetic acid.

Stainless steel surface: 54 mm diameter glass tubes containing a stainless-steel piece (1 cm × 2 cm × 1 mm) were inoculated with 5 mL of a 1:50 dilute culture and incubated at 37 °C for 72 h. The piece of stainless steel was then taken out aseptically, dried, and transferred into a sterile empty tube. Steel pieces were immersed in phage (1 × 10^5^ PFU/mL) for 10 min. Steel pieces were then taken out into sterile empty tubes and incubated at 37 °C for 24 h. Crystal violet bio-film assay was performed by adding 2 mL volumes of 0.1% crystal violet tube; the piece of steel was then transferred to a fresh tube and then washed with 2 mL of 30% acetic acid. The absorbance of the resultant solution was recorded.

Since the three materials’ surface areas differed, the absorbance values were normalized and calculated per square millimeter to compare the biofilm formation and dispersion achieved on the three surfaces.

The percentage of biofilm dispersed was calculated as follows:$$\mathrm{Percentage}\;\mathrm{biofilm}\;\mathrm{dispersed}=\frac{\mathrm{Absorbance}\;\mathrm{of}\;\mathrm{cell}\;\mathrm{control}-\mathrm{Absorbance}\;\mathrm{of}\;\mathrm{phage}\;\mathrm{treatment}}{\mathrm{Absorbance}\;\mathrm{of}\;\mathrm{cell}\;\mathrm{control}}\times 100$$

## 3. Results

### 3.1. Characteristics of Bacterial Isolates

Out of 67 wound swabs collected from patients, 35 swabs (16 DW and 19 from NDW) yielded Gram-negative bacilli that were catalase and oxidase positive, oxidative, and citrate-positive, with pigmentation on cetrimide agar, thus confirming them as members of the genera *Pseudomonas*. All isolates were positive for the presence of the *P. aeruginosa* species-specific gene *oprL*. A total of 10 clinical isolates and all 19 environmental isolates were sensitive to all antibiotics tested, while 25 clinical isolates were resistant to at least 1 antibiotic. Isolates from diabetic wounds were relatively more resistant to cefoperazone, piperacillin, levofloxacin, and gentamicin; the isolates from non-diabetic wounds were relatively more resistant to imipenem, aztreonam, and piperacillin/tazobactam. Isolates ADWS 10 and ADWS11 (DW isolates) and ADWS16, ADWS25, ADWS40, ADWS41, and ADWS53 (NDW isolates) were multidrug-resistant based on the susceptibility patterns observed (Appendix A).

### 3.2. Characterization of the Bacteriophages

Fourteen plaques from different water samples were isolated by the soft agar overlay technique. The lysates were purified by single-plaque purification (Figure 1a–c). The host range of the 14 phages was determined on the 35 clinical isolates and 19 environmental isolates. The electron micrograph of three phages, namely Pa3, Pa10, and Pa18 (chosen based on host spectrum and clear plaque morphology), showed the presence of tailed phages belonging to the family Myoviridae (Figure 1d–f).

### 3.3. Lytic Activity of Phages

The lysates were grouped into nine different phages designated Pa1, Pa3, Pa6, Pa7, Pa9, Pa10, Pa14, Pa15, and Pa18 based on the host range and similarity in plaque morphology. Isolates ADWS44, ADWS47, ADWS55, and ADWS58 from diabetic wounds and ADWS52 from a non-diabetic wound were not sensitive to any of the phages tested. All the multidrug-resistant isolates were sensitive to phages. Phage Pa3 showed a broad host range covering 46 of 55 isolates (~84% coverage) followed by phage Pa10 that covered 39 of 55 isolates (71%). In combination, nine phages covered 91% of the hosts (Figure 1g).

Phages Pa3, Pa10, and Pa18 were selected for the viable count reduction assay. The viable counts of PAO1 recovered after phage infection were plotted versus time and are represented in Figure 2. The survivors that regrew after initial phage exposure retained their parent host phage susceptibility, i.e., they were sensitive to all phages used in this study after two rounds of sub-culturing, indicating that the survivors were not phage resistors.

### 3.4. Antibiofilm Activity of Phages

Isolates were categorized as strong, moderate, and weak biofilm producers per established protocols [24]. All clinical isolates, irrespective of source, were strong biofilm formers. There was no significant difference between the biofilm-forming ability of the isolates from the two clinical sample types, i.e., DW and NDW. Environmental isolates AMO1, SO2, JMO2, and VM02 were moderate, while the other 15 were strong biofilm producers. 

Considering the ability of these phages to reduce/remove biofilms: the nine phages were able to remove 58% of the biofilm (mean of values) of DW isolates and ~56% of the biofilm of isolates from non-diabetic wounds. The efficiency of the antibiofilm activity of the phages on environmental isolates was an average of 84% (Figure 3 and Appendix A). The phages tested in this study could remove 75% of the biofilm in NDW isolates ADWS4, ADWS15, ADWS21, and DW isolate ADWS17 [ANOVA: F (1, 34) = 25.31; *p* = 0.0000157]. In the case of DW isolates, biofilms of isolates ADWS10, ADWS28, and ADWS46 that were lysed in broth were relatively resistant to phage treatment (<30% reduction), while among the NDW, ADWS34 and ADWS54 biofilms did not get effectively removed (<25% biofilm removed) by phage treatment. While the biofilms of most isolates showed varied susceptibility to removal by phages (>5% deviation from the mean), the biofilm formed by DW isolate ADWS06 was equally sensitive to all nine phages.

### 3.5. Activity of Phages on Biofilms Formed on Different Materials

Biofilms of PAO1 on polystyrene, glass, and stainless steel were grown, and the effectiveness of the isolated phages in dispersing biofilms on these surfaces was studied. There was no significant difference in the amount of PAO1 biofilm formed across the three surfaces (one-way ANOVA: F (2, 9) = 1.64, *p* = 0.23). All nine phages tested reduced the biofilm though at varied levels: Phages Pa1, Pa6, Pa15, and Pa14 were able to disperse more than 90% of the biofilm on the polystyrene surface, and phages Pa3, Pa15, Pa14, and Pa10 were efficient in dispersing more than 90% of the biofilm formed on the stainless steel surface. The maximum dispersion observed on the glass surface was 80% by phage Pa10. There was no significant difference in the ability of the phages to disperse PAO1 biofilms on a particular surface when compared with each other (F (9, 2) = 1.59, *p* = 0.189). However, there was a significant difference in the ability of phages to disperse PAO1 biofilms formed on the different surfaces tested (F (2, 9) = 16.06, *p* = 0.00009). Better dispersion was observed on stainless steel and polystyrene than the borosilicate glass (Figure 4). 

## 4. Discussion

*P. aeruginosa* is considered to be one of the important pathogens causing surgical site and wound infections. Drug-resistant *P. aeruginosa* and the implications thereof have been reported by various researchers [4,5]. While for certain pathogens, such as *E. coli*, the pathogenic strains are very distinct from commensal and environmental strains [25]. Clinical and environmental isolates of *P. aeruginosa* show little genetic variability and indicate that the environmental strains that can produce virulence-associated traits become pathogenic [17]. Based on this, we hypothesized that the phage susceptibility of environmental and clinical isolates should also be similar. The study approach aligns with the recommendations of several authors who have emphasized using clinical and environmental isolates instead of laboratory strains to generate more reliable data and a realistic assessment. Enhancing the applicability of the study and providing insight into the dynamics of phage-bacterial interactions in different environments. 

The isolates recovered in the study were characterized based on biochemical tests, confirmed as *P. aeruginosa* based on the presence of the *oprL* gene, and grouped based on source and antimicrobial susceptibility. *P. aeruginosa* isolates were resistant to several anti-pseudomonal drugs, and 4 of the 35 clinical isolates (11.4%) were multidrug resistant. The resistance of clinical isolates towards aztreonam and cefoperazone, the commonly used antibiotics, is similar to other studies reported from India [26,27,28,29]. The recovery of antibiotic-susceptible isolates from the environment reiterates that the pressure rendered due to antibiotics in a healthcare set-up favors the emergence of resistant forms. Using molecular typing techniques to determine the phylogeny of these strains would provide a more comprehensive understanding of the relationships between antimicrobial resistance and phage sensitivity of the different strains. However, previous studies using random amplified polymorphic DNA analysis (RAPD) as a molecular typing technique showed no distinct lineages of wound and sputum isolates [30]. 

Phage therapy is the most viable solution for combating the current multidrug resistance scenario. However, the ability of phages to act on the biofilms of these pathogens is crucial to its application. From an initial 14 isolated plaques, 9 different phage groups were identified. The difference in plaque morphology and host range are used as the primary tools for phage classification [31,32,33]. The isolated phages had a 91% host coverage spectrum among the isolates tested. The ability of these phages to act on multidrug-resistant isolates is a promising approach for phage therapy. While all the isolates from the environment were sensitive to at least one isolated phage, four isolates from diabetic wounds and one from non-diabetic wounds were resistant to all the phages tested. However, it is noteworthy that these phage-resistant isolates were susceptible to the antibiotics tested. NDW isolates were relatively more sensitive to many phages than DW. The susceptibility of environmental isolates to multiple phages was comparatively low. Three phages, Pa3, Pa10, and Pa18, were selected as candidate phages for electron microscopy to determine the phage structure and the effective MOI for reducing viable planktonic cells by the CFU reduction assay.

Phage morphology, as observed by electron microscopy, has been the most critical factor for phage taxonomy. However, the current system of virus nomenclature is genome based to provide a coherent and unified system of virus classification [34]. The three phages belong to the family Myoviridae based on the morphology-based taxonomy. Tailed phages constitute more than 99% of reported lytic phages; our observations agree with this. Whole phage genome sequencing and annotation would provide a better insight into the nature of these phages and probable answers to the difference in their activities. 

Biofilm formation is a severe problem in the health sector and has become one of the main culprits for initiating antibiotic resistance in bacteria. Approximately 80% of infectious and persistent bacteria can form biofilms. The ability of *Pseudomonas* spp. to construct biofilms on biotic and abiotic surfaces, such as glass, plastic, wooden pieces, clinical instruments, and even on stainless steel surfaces, differentiates them from other genera. *P. aeruginosa* forms biofilms on implanted biomaterials within hospital surfaces and water supplies. Phage and phage-derived enzymes are promising agents for dispersing bacterial biofilms [35]. The phages did not disperse biofilms of isolates resistant to the phage by routine test dilution. Sillankrova et al. reported the activity of four *Pseudomonas* phages, of which only one effectively reduced biofilms [36]. In our study, the three candidate phages could effectively reduce planktonic cells at MOI lesser than 1 by the CFU reduction assay. The nine phages could remove biofilms, though the biofilm reduction was not >99.9%. All the isolated phages could effectively reduce the biofilm formed by clinical and environmental isolates. Biofilms of 4 out of 12 isolates from diabetic wounds that were sensitive to phages by spot assay resisted removal by phage treatment, while among the non-diabetic isolates, only 2 of 18 biofilms resisted phage removal. All environmental isolates that were sensitive to phages by the spot assay were sensitive to phages for biofilm removal. Thus, the isolates from diabetic wounds appear more resistant to antibiotics and phage treatment. The number of isolates included in this study is very few to extrapolate this data for a population but indicates the differences in virulence and susceptibility of isolates obtained from different clinical sites [37]. Since biofilms impede the penetration of antibiotics, a disruption in the matrix would be sufficient to aid antimicrobial penetration. Additionally, repeated phage administration, an accepted therapeutic practice, was not tested, which is a limitation of this study.

Biomass removal depends on the biofilm age, the conditions under which the biofilm formed, and the types of phages applied. Phages can disrupt the biofilm matrix with or without EPS depolymerase, but depolymerase increases the phage’s degradative properties [38]. The phage genomes would provide insights into the genome structure, replication mechanisms, presence of depolymerases, or other exopolysaccharide-degrading enzymes and critical genes involved in host interaction, which could further empower unraveling the differences in the antibiofilm activities seen among the phages. In a study on *P. aeruginosa* isolates from chronic rhinosinusitis isolates, a cocktail of four bacteriophages decreased the biofilm biomass by a median of 76% at 48 h. Similarly, while phages individually reduced carbapenem-resistant *P. aeruginosa* biofilms by 50%, when used as a cocktail, the antibiofilm property increased to 72.9% [39]. Our study showed that, individually, phages could decrease biofilms by a median of 60% at 24 h.

A cocktail of phages may have provided a better overall reduction percentage. Latz et al. reported the activity of three phages on biofilms of *P. aeruginosa* and attributed the biofilm-degrading activity to smaller-size phages. They have also suggested that phages may not have an excellent effect on high-density biofilms but can prevent further accumulation and diffusion of biofilms by reducing migratory bacteria [40]. There was no significant difference in the ability of phages to reduce biofilms, indicating that the phages may not differ much in size. The electron micrographs of the three candidate phages concur with this. Studies on the ability of phages to reduce *Pseudomonas* biofilms on endotracheal tubes showed a ~50% reduction in optical density when 10^8^ bacteriophages were added and incubated for 24 h [41]. Phages are known to act better and for prolonged periods when added at low numbers to grow and lyse more bacterial cells in successive generations. Our study used 1000 PFU/mL, which shows comparable activity to higher concentrations. A *Pseudomonas* phage capable of killing planktonic cells and biofilm after 6 h of treatment has been reported [32]. In our study, the three phages tested were able to reduce planktonic cells within 2 h of treatment and sustained low counts for up to 10 h, while 50% biofilm reduction was sustained up to 24 h. The ability of these phages to reduce biofilms indicates that their antibiofilm ability may be due to the presence of depolymerases. 

*P. aeruginosa* is ubiquitously present in healthcare settings. This species can use simple organic molecules as carbon and energy sources, thus enabling multiplication in solutions that would otherwise be incapable of sustaining high bacterial growth, such as mild antiseptics, saline solutions, and soaps. In these conditions, the bacteria can readily adhere to wet surfaces or in contact with liquids and thus form biofilms [42]. These inanimate objects can thus be a hiding niche for this dreaded pathogen. The antibiofilm activity of phages was tested on three surfaces: stainless steel, glass, and PES, all commonly encountered in a healthcare setting. While the bacteria efficiently formed biofilms on all these surfaces with no significant changes in the quantity of biofilm, phages were able to remove the biofilms from all surfaces. However, the antibiofilm activity on stainless steel and PES was significantly better than on glass. Strain differences and environmental factors, including the attachment surface, greatly influence the secretion of exopolysaccharides that form the biofilm matrix [43,44]. Psl, Pel, alginate, and polyhydroxyalkanoate are the key exopolysaccharides secreted by *P. aeruginosa* in biofilm formation and architecture [45]. The study on the activity of phages on biofilms produced on different surfaces by a single isolate was conducted and required further validation on a larger cohort of isolates. Evaluating the differences in the *P. aeruginosa* biofilm matrix composition formed on different surfaces and the hydrolytic activity of phages on each of these specific components would provide insights into the differences in phage antibiofilm activity. Molecular typing to ascertain the genetic diversity among the isolates could further increase the understanding of the relative differences in phage and antibiotic susceptibility observed in the bacterial isolates from different clinical sources.

Our observations that phages can disperse biofilms efficiently indicate that the phages have efficient mechanisms that can disrupt biofilms. The consortium created is effective on clinical and environmental isolates.

## 5. Conclusions

Bacterial isolates of *P. aeruginosa* from clinical sources are more resistant to antibiotics and can form robust biofilms compared to environmental isolates. The environment harbors many phages specific to clinical and environmental isolates. Clinical isolates are susceptible to phage infection in planktonic form compared to environmental isolates. Biofilms of *P. aeruginosa* isolated from diabetic and non-diabetic wounds resist removal by phages compared to biofilms formed by environmental isolates. However, this partial eradication of biofilm by phages can aid in complementing antibiotics that are unable to penetrate biofilms in a clinical set-up [46,47,48,49,50].

## Figures and Tables

**Figure 1 microorganisms-11-02230-f001:**
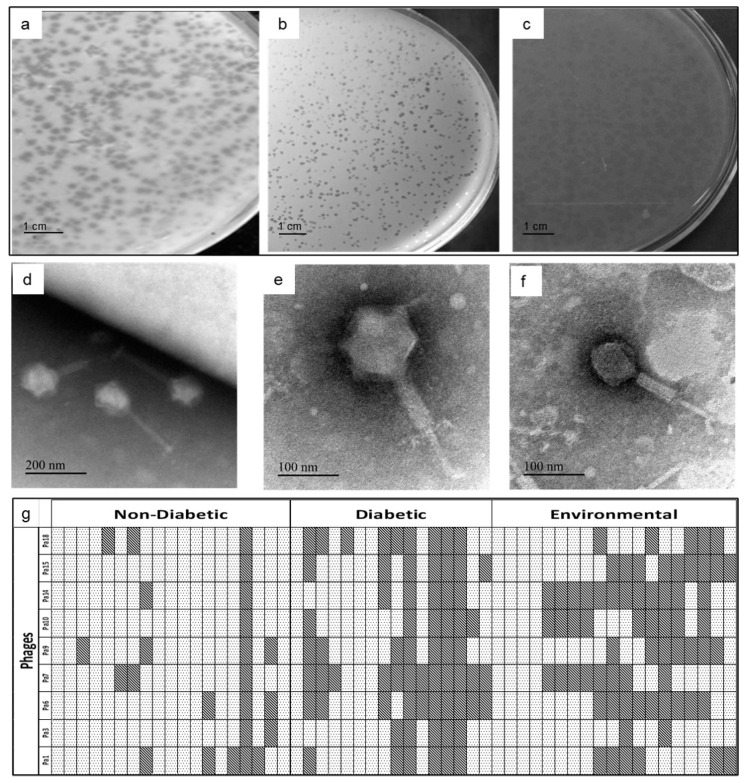
Representation of phage characterization: (**a**–**c**): Plaque morphology of phages Pa18, Pa10, and Pa3; (**d**–**f**): Transmission electron micrographs of Pa18, Pa10, and Pa3, respectively. All phages showed an icosahedral head, a distinct collar, a contractile tail, and a well-defined tail plate, conferring them to the family Myoviridae. (**g**) Host range: Rows represent phages Pa1, Pa3, Pa6, Pa7, Pa9, Pa10, Pa14, Pa15, and Pa18, respectively, and columns represent isolates. 
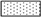
 indicates that the host is susceptible to that phage, while 
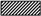
 represents resistance.

**Figure 2 microorganisms-11-02230-f002:**
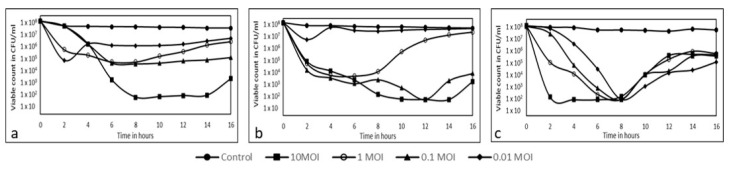
Viable count reduction: PAO1 treated with different concentrations of (**a**) phage Pa3, (**b**) Pa10, and (**c**) Pa18. Phage Pa3 at 0.01 MOI showed a significant decrease in cell count after 6 h up to 14 h [*p* = 0.000182]. 0.1 MOI and 1 MOI showed a >99.9% decrease in viable cell counts [*p* = 0.0053], while 10 MOI was inefficient in reducing bacterial counts [*p* = 0.330]. Phage Pa10 was ineffective in reducing viable count at 10 MOI [*p* = 0.723]. At 1 MOI though the cell counts were initially reduced by more than 2 log [*p* = 0.072], they gradually increased after 8 h. The lower MOIs of 0.1 and 0.01 effectively maintained low CFU recovery for up to 16 h [*p* = 0.000153 and 0.00023, respectively]. Phage Pa18 was highly efficient in reducing the bacterial viable counts [*p* < 0.001] for all MOI tested up to 8 h.

**Figure 3 microorganisms-11-02230-f003:**
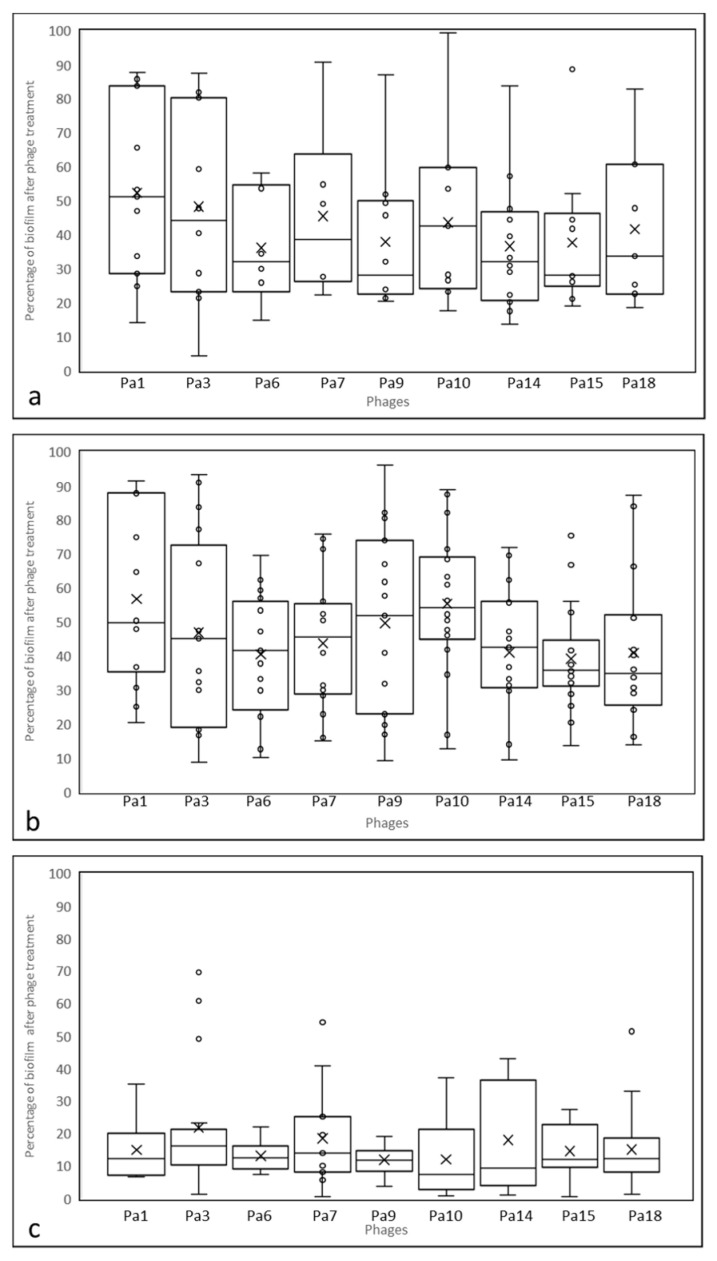
Ability of the phages to reduce biofilms in: (**a**) Isolates from diabetic wounds, (**b**) isolates from non-diabetic wounds, and (**c**) environmental isolates. Phages reduced ~50% of the biofilm of isolates from clinical sites while more than 80% of the biofilm of the environmental isolates. ^✕^ indicates mean; ^°^ represents individual values.

**Figure 4 microorganisms-11-02230-f004:**
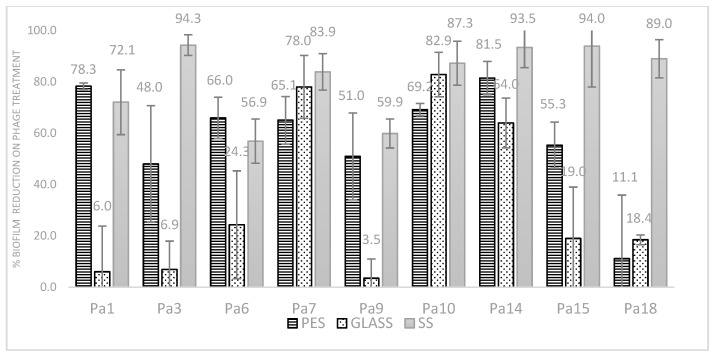
The activity of phages on PAO1 biofilms formed on different surfaces. The biofilm formed by the cell control was set as 0% dispersion. The relative difference between the cell control and phage treatment is percentage reduction—the higher the value, the greater the dispersion. PES—polystyrene, SS—Stainless steel.

## Data Availability

All data relevant to the study are included in the article. Any more supporting data may be available from the corresponding author on request.

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
