# Peer review of "Comparison of Antibiofilm Activity of Pseudomonas aeruginosa Phages on Isolates from Wounds of Diabetic and Non-Diabetic Patients"

_microorganisms, 2023, doi:10.3390/microorganisms11092230_

Round 1

Reviewer 1 Report

Microorganisms #2463936

Comments to the Authors;

This manuscript describes the effects of phages newly isolated on P. aeruginosa isolated from diabetic and non-diabetic wounds and environment. 

In this research, P. aeruginosa strains were isolated from diabetic and non-diabetic wounds and environments, characterized by the determination of antibiotic resistance and biofilm formation. On the other hand, authors isolated phages and also characterized them by checking the host ranges using the P. aeruginosa strains. Furthermore, the efficacy of the phages in P. aeruginosa biofilms using several strains was assessed. Additionally, the effects of phages on biofilms formed on three kinds of materials were investigated using only the PAO1 strain. The authors examined various characters. However, these results are not explained well-organized overall.

-Antibiotic resistances are essential to consider the prevalent P. aeruginosa strains. However, the importance of antibiotic resistance in the present study is not clearly described in the main text.  

- Sensitivities of P. aeruginosa isolated from diabetic wounds and environments showed relatively lower sensitivities than the strains isolated from non-diabetic wounds (Fig. 1g).  There were no differences between the biofilms formed by the strains DW and NDW (Page 6, Line 204.  No significant differences in the effects on the biofilms of them (Page 6, Line 208).

The reason why the authors compared the efficacies of phages to P. aeruginosa strains isolated from diabetic and non-diabetic wounds and environments is needed to describe clearly? 

-“However, this partial eradiation of biofilm by phages can aid in complementing antibiotics that are unable to penetrate biofilms in a clinical set up.” (Page10, Line 357

Data supporting this sentence is required.

Materials and Methods

Information is insufficient, especially in the Materials and Methods section.

Examples are indicated below.

Page 2, Line 80

What are the ”water samples”? Explain the details.

Page 2, Line 81

Clarify the “sample” by adding information.

Page 3, Line 110

Which is the correct incubation period for biofilm formation, five days or 72-h (described in the abstract)? 

How was the condition of the biofilm assay determined? 

Results

Page 5, Line 187, Fig. 2

Viable bacterial cells in the culture were reduced after the addition of phages. However, all three phages allow bacteria to grow later. The authors suggested that the bacterial cells in the culture (survivors) were not phage-resistant mutants because of the sensitivities to the phages determined by subculture. Without any data, it is not proven this description. 

Page4, Line 166

What is the purpose of determining the host range of 14 phages in this research? 

“chosen based on host spectrum ”

Does it mean that the host ranges of Pa3, Pa10, and Pa18 were much broader than those of others?

Headlines

Headlines in the Results section should be corrected appropriately to indicate the contents.

Minor comments

Page 3, line 103

It is better to show the values of MOI in the order from the minimum to maximum to avoid confusion. 

Fig. 1g

Universal design is currently recommended to adopt to the Figs.

It would be better to change the colors used in Fig. 1g.

Further improvement of vocabulary to choose the words generally used in scientific papers is recommended.

Reviewer 2 Report

Overall, the abstract provides a clear overview of the study's objective, methods, and key findings. It highlights the potential of bacteriophages as a strategy to combat biofilms and antimicrobial resistance. However, it is important to note that this review is based solely on the information provided in the abstract, and a more detailed analysis would require access to the full study.

The methodology section needs to be shortened as it includes an excessive amount of unnecessary details. It would be beneficial to focus on the key steps and provide a more concise overview of the experimental procedures.  It is necessary to use italics or an appropriate formatting style, such as italic font, when mentioning bacterial names. This will help to distinguish them from other text and maintain consistency throughout the article.  It is essential to consider an alternative classification of Pseudomonas bacteria in the context of bacteriophage infections. It might be worth exploring molecular typing techniques to determine the phylogeny of these strains, rather than solely relying on the source of isolation. This would provide a more comprehensive understanding of the relationships between different strains and their susceptibility to bacteriophages.  Regarding Figure 2, it is unclear whether the regrowing bacteria shown were resistant to bacteriophages. It would be beneficial to include information or discuss the susceptibility of these regrown bacteria to bacteriophage infection. This would help to interpret the significance of the regrowth observed.  The genetic characteristics of bacteriophages have been omitted from the discussion. It is important to include a section discussing the genetic aspects of bacteriophages, such as their genome structure, replication mechanisms, or key genes involved in host interaction. This will enhance the comprehensiveness of the study and provide a more holistic understanding of the bacteriophage-bacteria interactions.

Moderate editing of English language required

Round 2

Reviewer 1 Report

This manuscript is improved by correcting it according to the comments. However, there are descriptions still unclear. 

Line 22-23

 "Phages eradicated 85% of biofilms formed by environmental iso- lates compared to 58% biofilms of diabetic isolates and 56% biofilms of non-diabetic isolates."

Where is the DATA showing these results in the manuscript?

This sentence is confusable. Are these numbers the means of values analyzed? It is recommended to describe these results using the data (values) in the main text.

Line 43-44

 "Antimicrobial resistance (AMR) in P. aeruginosa especially carbapenem resistance has emerged as a global outbreak. "

This topic seems repeatedly mentioned in Line 324 with reference.

Line 89

The "orpL" (gene) but not "OprL" (gene product) is detected by PCR.

Line 114 and Line 132, line 164

5µL, 100µL, 2ml

 A blank space is needed between numbers and these units (5 µL, 100 µL, 2 ml).

Line 143

 "3 days. (72 hours)."

It should be described with any of "days" or "hrs."

Line 144

"air dried."

Why were the plates dried before the addition of phages? Bacterial cells or biofilm may be damaged before the addition of phages.

Line 145-146Line 161, Line 130-123 (Method)

What solution were the phages prepared in? Were phage lysates used in the experiments?

Line 153

 "Activity of phages on different materials"

The phages decrease the biofilm formed on the various materials. This sentence is still unclear. 

Line 188

 "*100" 

What does this asterisk?

Line 220, Line 247, line 272

The phages shown here are specific phages but not generalized "phages." It should be described clearly.

Line 338-344

"Three phages Pa3, Pa10 and Pa18 were selected as candidate phages for electron microscopy to determine phage structure and to determine the effective MOI for reduction of viable planktonic cells by the CFU reduction assay. The three phages belong to family Myoviridae based on their structure. as per the morphology-based taxonomy. Tailed phages constitute more than 99% of reported lytic phages and our observations are in consensus with this. The current system of virus nomenclature is genome based to provide a coherent and unified system of virus classification [34]. However, whole phage genome sequencing and annotation remains a cost intensive exercise in many countries."

The explanation about the present taxonomy, which was added from the previous version, should be moved before the description of classification by morphology.

This manuscript contains some inadequate descriptions, which may be confusable for readers.  The points required to be improved are indicated in the comments.

Author Response

Comment 1: Line 22-23

 "Phages eradicated 85% of biofilms formed by environmental isolates compared to 58% biofilms of diabetic isolates and 56% biofilms of non-diabetic isolates."

Where is the DATA showing these results in the manuscript?

This sentence is confusable. Are these numbers the means of values analyzed? It is recommended to describe these results using the data (values) in the main text.

 Response:  The data for these is now included in the manuscript and supporting information Table 2 provided.

Line 223 : Considering the ability of phages to reduce/ remove biofilms: the 9 phages were able to remove 58% biofilm (mean of values) of DW isolates and ~56% biofilms of isolates from non-diabetic wounds. The efficiency of antibiofilm activity of the phages on environmental isolates was an average of 84% (Figure 3 and supporting information Table S2).

Line 231: While the biofilms of most isolates showed varied susceptibility to removal by different phages (>5 % deviation from mean), the biofilm formed by DW isolate ADWS06 was equally sensitive to all the 9 phages.

Comment 2: Line 43-44

 "Antimicrobial resistance (AMR) in P. aeruginosa especially carbapenem resistance has emerged as a global outbreak. "

This topic seems repeatedly mentioned in Line 282 with reference.

 Response: The repeated sentence has been rephrased in line 282.

Line 284: Phage therapy appears to be the most viable solution for combating the current multidrug resistance scenario.

Comment 3: Line 89

The "orpL" (gene) but not "OprL" (gene product) is detected by PCR.

Response: We apologise for the error. It has been corrected.

Line 83: (PCR) for oprL

Line 171:P. aeruginosa species-specific gene oprL.

Line 272 : based on the presence of the oprL gene.

Comment 4: Line 114 and Line 132, line 164

5µL, 100µL, 2ml

 A blank space is needed between numbers and these units (5 µL, 100 µL, 2 ml).

Response: Spaces included between numbers and units throughout the text

Comment 5: Line 143

 "3 days. (72 hours)."

It should be described with any of "days" or "hrs."

Response:

Durations of incubation described as 72 hours in all instances.

Comment 6: Line 144

"air dried."

Why were the plates dried before the addition of phages? Bacterial cells or biofilm may be damaged before the addition of phages.

 Response:

The plates were air dried instead of washing repeatedly with PBS to ensure no planktonic cells remained in the wells. We have observed that washing with PBS can disrupt the biofilm due to forceful aspiration with a pipette. (Rai et al., J Health Allied Sci NU, 2021)

Comment 7:

Line 145-146、Line 161, Line 130-123 (Method)

What solution were the phages prepared in? Were phage lysates used in the experiments?

Response:

The phage preparations used were ultracentrifuged preparations and not lysates. The same has been clarified in the text.

Line 102: Fifty ml each of phage lysates were purified by ultracentrifugation at 32000 g for 4 hours. The pellet suspended in 500 µl SM buffer ( 50 mM Tris-HCl pH7. 5, 100 mM NaCl, 8 mM MgSO4, 0.01% Gelatin). These purified phages were diluted in SM buffer as per required number of phages and used for all further studies.

Comment 8: Line 153

 "Activity of phages on different materials"

The phages decrease the biofilm formed on the various materials. This sentence is still unclear. 

Response: Statements reframed to give better clarity on the content.

Line 137: Activity of phages on biofilms formed on different materials

Line 239: Activity of phages on biofilms formed on different materials

Comment 9: Line 188

 "*100" 

What does this asterisk?

Response: The * is the mathematical function of multiplication. This has now been replaced with “x” in Line 134 and 161.

Comment 10: Line 220, Line 247, line 272

The phages shown here are specific phages but not generalized "phages." It should be described clearly.

Response: To specify that the observations are with respect to the phages isolated in this study and not a generalized statement, the sentences have been reframed.

Line 227 : The phages tested in this study

Line 241: … of the isolated phages to disperse biofilms on these surfaces was studied

Line 289: The isolated phages had a …

Line 317: The 9 phages could

Line 318: All the isolated phages

Comment 11: Line 338-344

"Three phages Pa3, Pa10 and Pa18 were selected as candidate phages for electron microscopy to determine phage structure and to determine the effective MOI for reduction of viable planktonic cells by the CFU reduction assay. The three phages belong to family Myoviridae based on their structure. as per the morphology-based taxonomy. Tailed phages constitute more than 99% of reported lytic phages and our observations are in consensus with this. The current system of virus nomenclature is genome based to provide a coherent and unified system of virus classification [34]. However, whole phage genome sequencing and annotation remains a cost intensive exercise in many countries."

The explanation about the present taxonomy, which was added from the previous version, should be moved before the description of classification by morphology.

Response: The section has been reframed as suggested

Line 346 : Phage morphology, as observed by electron microscopy, has been the most critical factor for phage taxonomy. However, the current system of virus nomenclature is genome based to provide a coherent and unified system of virus classification [34]. The three phages belong to the family Myoviridae based on the morphology-based taxonomy. Tailed phages constitute more than 99% of reported lytic phages; our observations agree with this. Whole phage genome sequencing and annotation would provide a better insight into the nature of these phages and probable answers to the difference in their activities.  

Comments on the Quality of English Language

This manuscript contains some inadequate descriptions, which may be confusable for readers.  The points required to be improved are indicated in the comments.

Response: Minor editing for improving the language of the text has been done.

Reviewer 2 Report

in figure 4 shortcuts PES, SS  should be explained in caption

 Minor editing of English language required

Author Response

Comment : In figure 4 shortcuts PES, SS  should be explained in caption

Response: Caption edited to include expansions

Line 256:  …..the greater the dispersion. PES – polystyrene, SS -Stainless steel.

Comments on the Quality of English Language   : Minor editing of English language required

Response: Minor editing for improving the language of the test has been done.

Round 3
